# Gender Differences in Soft Tissue and Bone Sarcoma: A Narrative Review

**DOI:** 10.3390/cancers16010201

**Published:** 2023-12-31

**Authors:** Ilaria Cosci, Paolo Del Fiore, Simone Mocellin, Alberto Ferlin

**Affiliations:** 1Veneto Institute of Oncology IOV-IRCCS, 35128 Padova, Italy; ilaria.cosci@iov.veneto.it; 2Soft-Tissue, Peritoneum and Melanoma Surgical Oncology Unit, Veneto Institute of Oncology IOV-IRCCS, 35128 Padua, Italy; simone.mocellin@iov.veneto.it; 3Department of Surgical, Oncological and Gastroenterological Sciences (DISCOG), University of Padua, 35128 Padova, Italy; 4Unit of Andrology and Reproductive Medicine, University Hospital of Padova, 35128 Padova, Italy; alberto.ferlin@unipd.it; 5Department of Medicine, University of Padova, 35128 Padova, Italy

**Keywords:** gender differences, sarcoma, soft tissue cancer

## Abstract

**Simple Summary:**

This review focusing on gender differences in the incidence of soft tissue and bone sarcomas. Sarcomas are rare cancers arising from mesenchymal tissues, which are different from the epithelial tissues and originate from the embryonic mesodermal layer. These cancers can be classified into bone or soft tissue sarcomas. Most sarcomas occur without known causes; however, certain genetic syndromes and environmental factors are known to be associated with these malignancies. Studies have indicated a male predominance in sarcoma incidence, which is also seen in other cancers like colorectal and lung cancers. Notably, childhood sarcomas exhibit significant gender differences, with a stronger association with the male sex, particularly in soft tissue sarcomas. The biological reasons for these sex differences are not well understood, and this review seeks to shed light on these underlying factors to aid in prevention and treatment strategies.

**Abstract:**

Sarcomas, uncommon malignancies, stem from mesenchymal tissues, distinct from epithelial tissues, originating in the embryonic mesodermal layer. These sarcomas have been categorized as either bone or soft tissue sarcomas, depending on their originating tissue. The majority of sarcomas occur sporadically with their etiology being unknown, but there are several, well-established genetic predisposition syndromes and some environmental exposures associated with specific sarcomas. Recently, many studies have shown that sarcomas, in analogy with colorectal, skin, head and neck, esophageal, lung, and liver carcinomas, also have a male sex predilection. Significant gender differences have already been observed in childhood sarcomas. Among the tumors strongly associated with the male sex, childhood sarcomas have been identified as being particularly sensitive to the biological differences between the sexes, with special regard to soft tissue sarcomas. As the biological mechanisms underlying the sex differences in the incidence of soft tissue sarcomas remain largely unexplored, this review aims to highlight the factors underlying these differences to inform prevention and treatment.

## 1. Introduction

Sarcomas are tumors of mesenchymal origin, accounting for approximately 15% of all cancers in children and 1% of all malignancies in adults [1]. According to histopathological criteria and the primary site of occurrence within tissues, the World Health Organization has delineated over 70 distinctive histological subtypes of soft tissue sarcomas (the most common of which are shown in Figure 1) and about 10 subtypes of bone sarcomas (the most common of which are shown in Figure 2) [2].

Sarcomas are rare tumors (their annual incidence is lower than six new cases per 100,000 people) that can occur in virtually any part of the body. Their prognosis varies greatly across different histological subtypes and depends also on the stage the tumor is diagnosed. Due to their rarity and heterogeneity, sarcomas show better outcomes when treated within the frame of specialized centers, with inappropriate medical management being reported in more than 70% of patients treated outside dedicated centers [3,4].

Although most sarcomas typically occur without a known cause, some types are attributable to various confirmed genetic predisposition syndromes and specific environmental factors. However, although the etiology of sarcomas is not yet clear, numerous studies agree that sarcomas, like other types of cancer known so far, arise predominantly in men. This gender disparity has already been observed in childhood sarcomas, where there is a strong association between the male sex and soft tissue sarcomas. Because the precise biological mechanisms responsible for these sex-based differences in sarcoma incidence remain largely unexplored, this review seeks to shed light on the factors that underlie these distinctions, improving our understanding for prevention and treatment.

### 1.1. Environmental Factors and Sarcomas

Although the etiology of sarcomas remains poorly understood, several environmental and genetic factors have been identified that are responsible for their development.

Previous studies have shown that exposure to radiation for the treatment of other cancers, such as lymphoma, breast, testicular, ovarian, prostate, and lung, increases the risk of developing sarcomas. While radiation-associated sarcomas (RAS) are typically infrequent, affecting less than 1% of individuals undergoing radiation therapy, their occurrence is anticipated to rise due to the expanded utilization of radiation therapy for specific tumor types and the overall enhancement in cancer patient survival rates.

Radiation-associated sarcoma is a rare iatrogenic malignancy that occurs after radiotherapy for a high malignant grade and accounts for approximately 0.5–5.5% of all sarcomas. This event appears to be due to the ability that ionizing radiation has to damage the DNA triggering a spectrum of molecular changes, ranging from minor single base substitutions to severe genomic disorders. The use of whole genome sequencing in a limited subgroup of radiation-related neoplasms (carcinomas and sarcomas) has allowed the identification of particular traits highlighting a greater prevalence of small deletions (with a higher ratio to insertions) compared to non-radiation-induced tumors.

Furthermore, these neoplasms also show balanced inversions responsible for chromotripsy and other structural abnormalities damage [5].

The 3-year survival rate for individuals with radiation-associated sarcoma varies and is generally considered poor due to the aggressive nature of the disease and its resistance to chemotherapy. A study by Xi et al. reported that the 3-year overall survival rate was 32.4% among treated patients with RAS, with the median survival being 21.2 months. Complete surgical resection was identified as a major prognostic factor for survival. Another study by Wei et al. reported a 3-year overall survival rate of 19.1%, with a 3-year survival rate with no disease at 11.1% [6,7].

In a recent retrospective study focusing on radiation-induced sarcomas, it was found that the overall median survival for RAS patients at 3 years was 36 months.

This study, however, noted no significant survival differences when stratifying patients by various factors such as age at radiation therapy, latency time, and age at RAS occurrence. This indicates that these factors may not have a straightforward impact on the survival of RAS patients [5].

While the 5-year survival rate for individuals with RAS ranges from 17% to 58%, it is considerably lower than the 54% to 76% rate observed in the cases of sporadic sarcomas [8].

A study comparing radiation-associated sarcoma of the pelvis/sacrum (RASB) and primary osteosarcoma/spindle cell sarcoma of pelvis/sacrum (POPS) found that older RASB patients were less likely to receive chemotherapy and more likely to have higher perioperative mortality and worse 5-year disease-specific survival. No difference was noted in local recurrence or metastasis-free survival at 5 years. Overall, RASB shows poorer outcomes compared to POPS [9].

Diagnostic protocols for RAS continue to follow guidelines established in the 1940s, which have undergone few changes over time. These guidelines are based on three fundamental principles: the documented history of radiation exposure preceding the development of the sarcoma, the onset of the sarcoma in or near the radiation-exposed area, and the confirmation of the sarcoma through a histological diagnosis and histological diagnosis of sarcoma other than cancer in the first instance [10].

Although ionizing radiation remains the most relevant environmental factor associated with the development of sarcoma, other environmental elements have been investigated. In particular, increased incidence and mortality for some types of cancer, particularly soft tissue sarcomas (STS), have been observed in individuals exposed to both agricultural and non-agricultural chemicals. The link between these chemicals and STS was initially highlighted by Hardell et al. in 1977 [11]. This evidence was supported in 1995 by the same authors in a Swedish case–control study conducted in the 1970s and 1980s, which confirmed a significantly increased risk of STS with exposure to these compounds, with an odds ratio (OR) of 2.7 (95% confidence interval (CI) 1.9, 4.7) and OR of 3.3 (95% CI: 1.8, 6.1) obtained for phenoxy acetic acids and chlorophenols, respectively [12].

A recent systematic review that included a total of 50 publications and 35 meta-analyses highlighted, in 16 studies involving 2254 participants, a combined odds ratio (OR) for sarcoma of 1.85 (95% CI: 1, 22, 2,82) about exposure to phenoxy herbicides and chlorophenols, with a pooled standardized mortality ratio based on four cohort studies with 59,289 participants of 40.93 (95% CI: 2.19, 765.90) [13]. Furthermore, exposure to vinyl chloride monomers resulted in pooled hazard ratios of 19.23 (95% CI: 2.03, 182.46) for hepatic angiosarcoma and 2.23 (95% CI: 1.55, 3, 22) for other STS, respectively, in three cohort studies involving 12,816 participants [14], while, in four cohort studies including 30,797 participants, an association between dioxin exposure and increased STS mortality was observed, showing a combined standardized mortality ratio of 2.56 (95% CI: 1.60, 4.10). Finally, woodworking occupations also represent an increased risk of developing STS, presenting an aggregate odds ratio of 2.16 (95% CI: 1.39, 3.36) [15].

### 1.2. Genetic Susceptibility and Sarcoma

Genetically, sarcomas are a type of tumor that can be divided into two groups. One group is characterized by simple karyotype defects consisting of disease-specific chromosomal translocations leading to abnormal gene (and protein) function, such as Ewing sarcoma, alveolar rhabdomyosarcoma, and synovial sarcoma. Furthermore, there are instances where complex karyotypes are observed, signifying a notable disturbance in genomic stability, seen for example in leiomyosarcoma, liposarcoma, undifferentiated pleomorphic sarcoma, osteosarcoma, angiosarcoma, and malignant peripheral nerve sheath tumor [16].

In addition, it is now known that individuals with certain genetic syndromes are predisposed to developing sarcoma. In particular, many inherited syndromes such as Familial gastrointestinal stromal tumor syndrome (GIST), Li–Fraumeni syndrome (LFS), neurofibromatosis (NF1), and retinoblastoma (Rb), Bloom syndrome (BS), fumarate hydratase (FH), Rothmund–Thompson syndrome (RTS), and Werner syndrome (WS) are responsible for an increased risk of developing this type of cancer [17].

Familial gastrointestinal stromal tumor syndrome (GIST)

Familial gastrointestinal stromal tumor (GIST) syndrome is a condition associated with sarcoma development, originating from the interstitial cells of Cajal in the gastrointestinal tract. Recognized recently as distinct, GISTs were previously classified as leiomyomas or leiomyosarcomas. Most GISTs (75% to 80%) exhibit c-kit gene mutations, especially in exon 11, which lead to increased function of a tyrosine kinase receptor, fostering cell proliferation. Additionally, mutations in the PDGFRA gene account for 5% to 15% of GIST cases [18]. Moreover, the syndrome is characterized by activating mutations in the KIT or PDGFRA genes, which include inherited mutations in specific exons of these genes. These mutations lead to continuous kinase activation, promoting tumorigenesis. Individuals with these mutations may develop multiple GISTs in the stomach or bowel [18].

Li–Fraumeni syndrome (LFS)

Li–Fraumeni syndrome (LFS) was initially identified and documented by Li and Fraumeni, along with their colleagues, after examining four family cases wherein either siblings or first cousins manifested pediatric sarcoma, while a parent had experienced early-onset cancer.

The Li–Fraumeni syndrome definition requires “the presence of a proband diagnosed with sarcoma before age 45, a first-degree relative younger than age 45 with any cancer, and a first- or second-degree relative younger than age 45 with any cancer or a sarcoma at any age” [19]. In 2009, a version of the Chompret Criteria was proposed to help clinicians to recognize Li–Fraumeni syndrome, expanding on the old definition and based on three criteria (Table 1) [20]. The syndrome is linked to TP53 gene mutations, which typically prevent tumor growth and promote cell death and DNA repair. These mutations often result in the loss of p53 function or impede the cell death pathway, contributing to sarcoma and other cancers due to genomic instability. Germline mutations in the p53 gene are found in Li–Fraumeni syndrome (LFS), with 30% to 60% of soft tissue sarcomas showing somatic p53 mutations. Affected individuals are prone to a range of tumors, including breast, brain, adrenocortical tumors, and various leukemias. Children with rhabdomyosarcoma under age 3 may particularly carry TP53 germline mutations [21]. A study from 2004 to 2015 with 89 identified carriers showed that those who underwent a comprehensive surveillance protocol for Li–Fraumeni syndrome had better outcomes. Surveillance led to the detection of 40 asymptomatic tumors and was associated with higher 5-year survival rates compared to those who declined surveillance. The study suggests incorporating surveillance into clinical management for improved survival in TP53 variant carriers [22].

Neurofibromatosis (NF1)

Neurofibromatosis is a genetic disorder that progresses slowly, primarily affecting neuroectodermal tissues like skin, nerves, bones, and eyes. It increases the risk of developing malignant peripheral nerve sheath tumors (MPNST), with up to a 10% lifetime risk. NF1, a subtype of this condition, involves a heterozygous mutation in the neurofibromin gene on chromosome 17q11.2. Neurofibromin is a tumor suppressor that regulates the Ras/MAPK/AP-1 pathway, and its loss leads to heightened Ras activity, which can promote tumor growth. NF1 is associated with various tumors, including neuroblastoma, neurofibroma, thymoma, and breast cancer [23].

Retinoblastoma (Rb)

Retinoblastoma is a childhood eye cancer arising in the retina, with hereditary and non-hereditary forms. Survivors of hereditary retinoblastoma are at increased risk of secondary tumors like osteosarcoma, and radiation treatment heightens this risk [24].

Retinoblastoma arises from germline mutations that deactivate one RB1 gene allele. Tumors form when both RB1 alleles mutate, followed by other genetic changes. Hereditary retinoblastoma patients are more likely to develop osteosarcoma, with RB1 mutations detected in 30–75% of these tumors. Hereditary survivors face a greater risk of secondary osteosarcomas than the general population or those with non-hereditary retinoblastoma [25,26]. RB1 gene mutations not only predispose individuals to retinoblastoma but also increase the risk for secondary tumors like soft tissue sarcomas, melanoma, brain tumors, and some carcinomas, including those of the lung, breast, and bladder [27].

Bloom syndrome (BS)

Bloom syndrome (BS) is a rare autosomal recessive disorder more common among Ashkenazi Jews, characterized by genetic instability and increased sarcoma risk. It is caused by mutations in the BLM gene on chromosome 15, which encodes a RecQ family DNA helicase essential for genomic stability. Ashkenazi Jewish BS patients often have a specific frameshift mutation. Cells from BS individuals show numerous cytogenetic abnormalities, such as increased chromosome breaks, quadriradial chromatid exchanges, and, notably, a high rate of sister chromatid exchanges (SCEs), which is a key diagnostic marker for the syndrome [28].

Rothmund–Thompson syndrome

Rothmund–Thomson syndrome is an autosomal recessive disorder caused by mutations in the RECQL4 gene, which increases cancer risk, particularly osteosarcoma. Mutations leading to RECQL4 loss-of-function include nonsense mutations, frameshifts, splice site changes, and intron deletions. Located on chromosome 8q24.3, RECQL4, a RecQ DNA helicase gene, is often amplified in osteosarcoma, near the MYC gene. Those with Rothmund-Thomson syndrome have a higher chance of developing malignancies, especially osteosarcoma [29].

Werner syndrome

Werner syndrome (WS) is a rare autosomal recessive disorder that emerges in the late teens or early twenties, simulating symptoms of accelerated aging like heart disease, osteoporosis, hair loss, cataracts, diabetes, and hypogonadism. It is caused by mutations in the WRN gene, a RecQ helicase on chromosome 8p11.1. People with WS have a higher risk for several cancers, including osteosarcoma, soft tissue sarcoma, meningioma, myeloid disorders, melanoma, thyroid carcinoma, and various epithelial cancers. There are over 90 known mutations affecting the WRN gene, including base substitutions, insertions, deletions, and complex mutations that disrupt its function and reading frames [30].

### 1.3. Sex Differences in Sarcomas

In recent years, epidemiological studies have highlighted sexual dimorphism as a relevant factor in the incidence and survival of many cancers, including colorectal, skin, head, neck, esophageal, lung, and liver cancers. Indeed, incidence rates ranging from 1.26:1 to 4.86:1 have been reported to be significantly higher in men than in women, irrespective of the ethnicity of the population studied [31,32,33,34].

Overall, the variations in question have commonly been linked to occupational or behavioral aspects. However, researchers have also explored cellular and molecular influences, particularly emphasizing the impact of sex hormones. Notably, these hormones might potentially influence cancer cells, elements within the tumor’s microenvironment, cellular metabolism, and the immune system [34].

Recently, Rong J et al. collected gastric GIST data from 2010 to 2016 through the SEER database, using, for the first time, propensity score matching (PSM) with a relatively large sample size. These authors aimed to investigate the relationship between sex and prognosis in patients with gastric GISTs using a method capable of reducing the influence of confounding factors.

The findings from this study emphasized the role of gender as a distinct factor influencing the prognosis of gastric GIST, revealing that males exhibit a heightened risk of mortality compared to females [35].

Indeed, compared to female gastric GIST, male patients were less likely to undergo surgical treatment (95.9% vs. 98.1%), more likely to have large tumors (>10.0 cm) (24.0% vs. 16.4%), and more likely to have a mitotic index greater than 10/50 HPF (14.1% vs. 9.7%). These data confirm the findings of previous studies reporting that the prognosis of gastric GIST is related to sex [36].

The link between variations in sex and prognosis might be attributed to the involvement of sex hormones. Evidence from other cancer studies indicates that the pathway associated with sex hormone signaling can impact susceptibility to cancer and the microenvironment of tumors, albeit operating through diverse mechanisms. These hormones play a role in controlling processes such as angiogenesis and inflammation, thereby affecting the progression of cancer differently among genders. For instance, there’s been a noted reduction in ERß levels within cancer scenarios.

A recent study has highlighted the interconnectedness of estrogen with the mucosal barrier, gastrointestinal functionality, and the regulation of intestinal inflammation. This hormone appears to play a protective role, particularly in gastrointestinal tumor development, notably in the case of colorectal cancer.

Indeed, it has been reported that the use of anti-estrogen drugs such as tamoxifen may increase the risk of gastric adenocarcinoma [37].

A prospective study conducted by Freedman ND and colleagues, as part of the Shanghai Women’s Health Study and based on population data, revealed an association between exposure to hormones like estrogen and a decreased likelihood of developing gastric cancer [38].

Additionally, exploring the notion that sex hormones might contribute to the development of esophageal or gastric adenocarcinoma, M Lindblad et al. conducted research on the protective effects of hormone replacement therapy (HRT) in women against these tumors. Their findings indicated that HRT usage correlates with a reduction of over 50% in the risk of gastric adenocarcinoma among users compared to non-users [39].

Newer studies have provided additional evidence supporting the notion that extended exposure to estrogen could potentially lower the chances of developing gastric cancer. Recent meta-analysis findings have revealed a noteworthy connection: the utilization of hormone replacement therapy (HRT), commonly employed to alleviate menopausal symptoms, was linked to a 28% decrease in gastric cancer risk compared to individuals not exposed to HRT. Moreover, a subgroup analysis focused on the type of HRT formulation demonstrated risk reductions in gastric cancer following the use of both estrogen-only therapy (pooled RR, 0.63; 95% CI: 0.51–0.77, I^2^ = 0%) and estrogen–progestin therapy (pooled RR, 0.70; 95% CI: 0.57–0.87; I^2^ = 0%) compared to non-users [39,40].

For soft tissue sarcomas (STS), the variability of incidence-based mortality by sex over the past decade has not been well studied, but many studies examining sex differences have highlighted that men have a higher incidence of STS compared with their female counterparts.

In England, an estimated 4295 cases of soft tissue sarcoma are diagnosed annually, representing a crude rate of 7.7 cases per 100,000 individuals (2017–2019). The distribution of soft tissue sarcoma cases in the UK appears to be relatively equal between females and males (1996–2010). Despite this equality, the incidence rates of soft tissue sarcoma, especially the European age-standardized (AS) rate, significantly differ between genders, being notably lower in females compared to males [33]. These results were confirmed by another analysis conducted by Hung and collaborators performed on the Taiwanese population according to the 2013 WHO classification.

The study highlighted a male predominance, particularly marked in Kaposi’s sarcoma (SIRR, 5.4; 95% CI: 4.41–6.63, *p* < 0.05), as well as in other subtypes such as UPS, liposarcoma, angiosarcoma, fibrosarcoma, GIST, chondrosarcoma (CS), and NOS sarcoma with a SIRR of 1.2–2.1 [41].

Hsieh et al. previously documented similar outcomes, exploring discrepancies in the occurrence rates and patterns of STS across different racial and ethnic groups in adolescents and young adults aged 15–29 years, considering sex, age, and histological type. Their findings revealed a 34% higher incidence of all STSs combined in males compared to females (95% CI: 1.28, 1.39), as well as a 60% higher incidence in Black individuals compared to Caucasians (95% CI: 1.52, 1.68) [42].

The demographic associations linked to STS in recent years and the probability of developing comorbidities have recently been analyzed to understand the mechanisms underlying the disease and to focus on diagnostic and therapeutic strategies.

In their examination of patients with STS, Van Herk-Sukel and colleagues discovered a notable escalation in the risk of developing medical complications, specifically highlighting a heightened risk for cardiovascular disease [43].

The proportion of STS in male and female patients may vary significantly depending on the type of tumor. A study analyzing the incidence and survival of cutaneous soft tissue sarcomas (CSTS) in the US population highlighted that CSTS rates varied markedly over time and by race, sex, and histological type, supporting the notion that the histological variants of CSTS are etiologically distinct and appear to affect males more than females [44]. Patel SJ et al., in a recent investigation utilizing data sourced from the National Cancer Institute’s Surveillance, Epidemiology, and End Results (SEER), analyzed mortality rates correlated with incidence between 2000 and 2016 across various categories such as tumor grades, gender, and racial demographics among individuals diagnosed with STS.

The authors confirmed that STS has a male sex predilection and that the male sex tends to have a higher incidence of mortality, regardless of tumor grade and race.

Indeed, this study highlights the higher incidence-based mortality rate in Caucasian males compared to African American males over the past 15 years, suggesting that soft tissue sarcomas in Caucasian males have worse outcomes [45].

As a result of these recent trends, the American Cancer Society predicts that 13,400 new soft tissue sarcomas will be diagnosed in the United States in 2023 (7400 in males and 6000 in females), with the majority of cases occurring in male patients. In addition, an estimated 5140 people (2720 males and 2420 females) will die from STS. Although the proportion of STS in male and female patients can vary significantly depending on the type of tumor, STS appears to affect males more than females [46].

As osteosarcoma is rarer than other sarcomas and is characterized by an incidence with a bimodal age distribution, little is known about its incidence in males and females.

Cole S et al. conducted a study utilizing data provided by the National Cancer Institute’s Surveillance, Epidemiology, and End Results (SEER) program to delve into younger osteosarcoma cases and examine racial minorities. Despite evidence suggesting a higher incidence of osteosarcoma among individuals of African descent, the authors extensively compared osteosarcoma occurrence and survival rates across four distinct age groups (0–9, 10–24, 25–59, and >60 years old) based on race/ethnicity, gender, decade, pathological subtype, and tumor location.

Their investigation uncovered a total of 2312 osteosarcoma cases within the 10–24 age category, nearly accounting for half of all osteosarcoma cases in the SEER 18 database. Additionally, it was noted that osteosarcoma demonstrated a higher prevalence in males compared to females across all race/ethnicity categories, displaying an overall male-to-female ratio of 1.3:1 (males, IR, 8.1; 95% CI: 7.7–8.6 vs. females, IR, 6.2; 95% CI: 5.8–6.6). Furthermore, in the >60 age group, males exhibited a higher prevalence of osteosarcoma compared to females, both overall (1.3:1) and within specific race/ethnicity categories [47].

Ewing sarcoma ranks as the second most prevalent bone sarcoma, displaying an incidence rate of 0.3 cases per 100,000 individuals annually. Within childhood cancer cases, this particular subtype of bone sarcoma constitutes approximately 2%. Notably, it exhibits a higher prevalence among males compared to females, maintaining a male-to-female ratio of roughly 1.5, and typically peaks in incidence around the age of 15 [48].

Gender differences in the incidence of radiation-associated sarcomas are not widely documented and the evidence may vary by study. However, some literature suggests that there could be a higher incidence in females, especially following radiation treatment for breast cancer, which is one of the more common scenarios leading to RIS. The specific incidence rates and gender differences can depend on a multitude of factors, including the type and location of the primary cancer treated with radiation, genetic predispositions, and the individual radiation doses administered. A recent study on RIS provided insights into the distinct genomic landscapes of these sarcomas. The research included 82 patients, predominantly females (83%), with a median age of 64 years. It compared radiation-associated angiosarcomas (RT-AS) with other RIS histotypes and sporadic sarcomas. The study found notable differences in the mutation and copy number alteration profiles among various RIS histotypes. RT-AS, especially those derived from breast radiation, had a unique genomic landscape with frequent MYC, FLT4, CRKL, HRAS, and KMT2D alterations. In contrast, other RIS types had genomic features similar to their non-radiation counterparts. The findings suggest that potential molecular targets for treatment could be specific to each histotype [49]. However, in the Inoue YZ study which evaluates the analysis of the clinical–pathological characteristics and treatment of patients with post-irradiation sarcoma of the bones and soft tissues, there does not seem to be a clear gender bias in this condition. Indeed, the ratio of males to females was 8:5, but this difference vanished when tumors specific to one gender (such as those affecting the breast, cervix, testis, or ovary) were excluded from the analysis. Additionally, no racial predilection has been noted in the research literature [50].

Furthermore, the onset of sarcomas associated with genetic syndromes does not seem to have a clear gender disparity. However, the manifestation of certain types of sarcomas may vary with age and the specific genetic mutation involved. For example, individuals with Li–Fraumeni syndrome (LFS), which is characterized by germline mutations in the TP53 gene, tend to develop sarcomas and other cancers at a younger age compared to the general population. Specifically, sarcomas represent a significant proportion of tumors in TP53 mutation carriers, with most occurring before the age of 50. The type of sarcoma can also be correlated with the type of TP53 mutation present, with certain mutations being associated with early-onset sarcomas like rhabdomyosarcoma in individuals younger than 5 years and osteosarcoma at any age [51].

It is important to note that knowledge of these genetic predispositions is crucial for the clinical management of sarcoma patients. Identifying individuals with heritable cancer predisposition syndromes can help tailor treatment strategies to minimize toxicity and maximize efficacy. Furthermore, understanding a patient’s genetic background can assist in the implementation of appropriate genetic counseling, as well as screening or surveillance strategies for both the patient and their relatives [52].

The research emphasizes that while treatment strategies for most sarcomas may not significantly differ between sporadic cases and those associated with predisposition syndromes, the recognition of genetic predisposition is vital for overall patient management [53].

### 1.4. Sex Differences in Sarcoma in Childhood

Significant sex differences have already been observed in childhood sarcomas. Among the tumors strongly associated with the male sex, childhood sarcomas have been identified as being particularly sensitive to sex biological differences.

Osteosarcoma (OS) ranks as the prevailing primary malignant bone tumor found among children and teenagers, with an estimated 4.8 new cases per million individuals under the age of 20 in the United States annually; this figure to roughly 450 cases per year in this age group, contributing to approximately 3% to 5% of childhood tumors. OS demonstrates a higher occurrence among males and African Americans. Following OS, the second most frequent primary malignant bone tumor among children and adolescents is Ewing sarcoma (ES), recording an estimated 2.9 new cases per million among individuals under the age of 20 in the United States annually.

Ewing sarcoma (ES) exhibits a slightly higher occurrence among males, and its frequency is nine-fold greater in Caucasians compared to African Americans [54].

The male predominance in childhood cancer incidence is well known, but few studies have focused on sex differences in incidence during childhood and adolescence.

Ognjanovic et al. conducted an analysis focusing on the incidence and survival patterns of rhabdomyosarcoma (RMS) among children and adolescents under the age of 20. This cancer type is typically categorized into two primary histological subtypes: embryonal RMS (ERMS) and alveolar RMS (ARMS), with ERMS accounting for 60–70% of cases and ARMS representing 20–30%. The investigation considered various demographic factors such as sex (male and female), age brackets (0–4 years, 5–9 years, 10–14 years, and 15–19 years), and racial backgrounds (white, black, American Indian/Alaskan Native, and Asian/Pacific Islander combined). In the nine SEER registers between 1975 and 2005 

A total of 987 cases of RMS were diagnosed among children aged 0–19 years. Males displayed a higher incidence of RMS compared to females, with rates of 5.2 per 1,000,000 and 3.8 per 1,000,000, respectively, resulting in a rate ratio of 1.37 (95% CI: 1.21–1.56). Notably, the male predominance in RMS was predominantly seen in ERMS, with a male-to-female rate ratio of 1.51 (95% CI: 1.27–1.80). These findings support previous reports indicating high incidence rates of RMS, its early onset before the age of 10 in more than 50% of cases, and a distinct male predominance [55].

This disparity has been found in most pediatric cancers, acute lymphoblastic leukemia, non-Hodgkin’s lymphoma, medulloblastoma, hepatic tumors, osteosarcoma, and germ cell tumors, showing that the direct effect of the male sex is significant for several tumor types. Furthermore the American Cancer Society reported a higher cancer incidence in males aged 0–14 years with a rate of 178.0 per 1,000,000, unlike females where a rate of 160.1 cases per 1,000,000 was observed, corresponding to a crude incidence rate ratio for childhood cancer for 1.11 for male versus female [56]. To shed light on potential mechanisms, such as hormonal fluctuations or periods of rapid growth, that may contribute to increased cancer incidence in males, Williams LA and collaborators used data from the Surveillance, Epidemiology, and End Results (SEER)’s 18 registries (2000–2015) to examine the association between male sex and childhood cancer by single year of age (0–19) and tumor type. The study found that male sex was positively associated with most cancers.

Particularly notable were the positive correlations found between the male gender and various types of cancers across different age groups. Neuroblastoma (IRR, 1.13; 95% CI: 1.07–1.19), retinoblastoma (IRR, 1.17; 95% CI: 1.08–1.26), and hepatoblastoma (IRR, 1.70; 95% CI: 1.53–1.86) exhibited significant associations with male sex across all ages. Notably, neuroblastoma showed a significant link with males at ages 1, 2 to 3, and 5, while retinoblastoma manifested this association at age 2. Hepatoblastoma, on the other hand, exhibited a notable correlation with males aged < 1 to 3. Furthermore, the male gender also displayed positive associations with nephroblastoma at ages < 1 to 1, and with various bone tumors (osteosarcoma IRR, 1.33; 95% CI: 1.25–1.41; chondrosarcoma IRR, 2.59; 95% CI: 2.08–3.10; Ewing sarcoma IRR, 1.69; 95% CI: 1.56–1.81), notably significant in osteosarcoma at ages 14 to 19. Ewing sarcoma consistently exhibited an association with male sex, significantly so from ages 9 to 19, except at age 1. “The Ewing sarcoma family” of tumors (Ewing tumors, Askin tumors, and pPNET) showed associations with male sex (IRR, 1.27; 1.06–1.48), similarly with rhabdomyosarcoma (IRR, 1.42; 95% CI: 1.33–1.51) and fibrosarcoma (IRR, 1.14; 95% CI: 1.00–1.28). Additionally, a strong association was observed between male sex and GCTs, particularly intracranial/intraspinal (IRR, 2.73; 95% CI: 2.51–2.95) and malignant gonadal GCTs (IRR, 2.35; 95% CI: 2.24–2.45) across all age groups. These higher incidence rates among males remained consistent from childhood through adolescence [57,58].

In a more recent analysis aiming to identify sex-based survival disparities in childhood cancers, the same researchers examined overall survival differences and estimated the risk of death in males versus females for 18 childhood cancers. This investigation utilized the Surveillance, Epidemiology, and End Results (SEER) program’s 18 registries spanning from 2000 to 2014.

This study not only confirmed that males had worse overall survival than females, but also reported worse survival and an increased risk of death for males diagnosed with ependymoma, neuroblastoma, and osteosarcoma. Furthermore, disparities in five-year survival rates based on gender were evident, indicating lower rates among males (85% for males, 88% for females). This trend was notably pronounced in specific cancers: ependymoma (71% male, 78% female), neuroblastoma (74% male, 78% female), hepatoblastoma (77% male, 82% female), osteosarcoma (64% male, 71% female), and Ewing sarcoma (67% male, 71% female). Notably, males exhibited significantly poorer overall survival across a 15-year observational period for ependymoma (log-rank *p* = 0.02), neuroblastoma (log-rank *p* = 0.003), and osteosarcoma (log-rank *p* = 0.004) [59]. However the concept of gender disparities in the incidence of certain sarcomas among different age groups remain an intriguing aspect of cancer epidemiology]. During childhood, particularly in the prepubescent years, some sarcomas show a marked difference in incidence between genders. For instance, embryonal rhabdomyosarcoma (ERMS) is more commonly diagnosed in boys than in girls [60]. The reasons behind this gender inequality in incidence rates are not fully understood but may involve genetic, environmental, and hormonal factors, even though these hormonal influences are not as pronounced before puberty.

As individuals age, the incidence of sarcomas still shows gender variations, but the reasons for these differences may shift. In older adults, hormonal differences between genders diminish, particularly post-menopause in women, when levels of hormones such as estrogen and progesterone decrease significantly. Despite the reduced hormonal differences, gender disparities in sarcoma incidence persist in the elderly. This could suggest that factors other than hormones, such as genetics, lifestyle, or environmental exposures, may play a more significant role in the development of sarcomas in older age groups [61].

### 1.5. Sex Biological Differences in Sarcoma in Childhood

The intricate biological pathways contributing to the variations in childhood cancer occurrence between genders have largely eluded extensive investigation.

However, it is thought that the increased cancer diagnoses in boys are mainly due to hormonal, genetic, and immune factors.

Some sex differences in cancer may be due to differences in the hormonal milieu. While this facet might not hold as much significance during early childhood compared to adolescence and adulthood, certain cancers like osteosarcoma, fueled by swift bone development and/or hormonal changes during puberty, could potentially be affected by fluctuations in hormones [62].

Williams LA et al. observed that osteosarcomas and Ewing sarcoma have a male-to-female ratio that fluctuates according to pubertal timing, which coincides with peak bone growth dependent on estrogen [58].

Recent research emphasizes the crucial involvement of the growth hormone (GH)/insulin-like growth factor 1 (IGF-1) axis in bone growth during puberty. Studies suggest that estrogen boosts GH secretion in both boys and girls, while testosterone primarily impacts GH secretion by converting it into estrogen through aromatization [63].

Differences in the onset of puberty and estrogen levels in males and females may regulate the rate of bone growth, which may explain the higher incidence of osteosarcoma and Ewing sarcoma in males during adolescence [47].

Furthermore, the strikingly higher male predominance observed in chondrosarcoma compared to other bone tumors raises noteworthy considerations, suggesting a potential involvement of hormonal or growth-related factors in this observed pattern. Population-based studies indicate a protective effect for women in cancer survival, potentially linked to hormonal differences. In high-grade CS, the female gender, likely influenced by estrogen, significantly improves survival compared to males. This effect is age-dependent, diminishing post-menopause. Estrogen’s role in bone and cartilage development and its presence in CS suggest a potential for anti-estrogen therapy. Despite inconsistent findings across studies, the age and gender impact on chondrosarcoma survival highlights the importance of further research into the interaction between sex hormones and high-grade CS [48,64].

The consistent prevalence of males in medulloblastoma is likely attributed to the asymmetric impact of sex on brain development. Animal studies have highlighted the contribution of prenatal hormones, sex chromosomes [65], and the immune system in influencing early neural sexual differentiation [66].

The elevated risk of childhood cancer among males may also be rooted in genetic factors. Although studies of sex-biased gene expression during human development are limited in adults, it has also been observed in children that genes on the X chromosome show higher variation in expression between the sexes than autosomal genes, and this may depend on sex differences in chromatin accessibility. Additional genetic mechanisms could involve somatic mutations occurring on the X or Y chromosome, affecting males and females differently. Over time, a preference toward one X chromosome may emerge, potentially confining mutations to the inactive copy due to selection pressure. In contrast, any mutation of the X chromosome in males is obligatory throughout their lifespan and could exert a more pronounced influence on gene expression and cancer development [67]. Given that the Y chromosome contains fewer than 100 genes, its role in increasing cancer risk in males cannot be ignored [68].

Finally, in both children and adults, disparities in cancer risk between the sexes may be linked to differences in the immune response to tumor development. Notably, males typically exhibit lower innate and adaptive immune responses than females, evidenced by higher rates of infectious diseases in males and elevated rates of autoimmune diseases in females.

This attribute likely arises from the abundance of immune-related genes located on the X chromosome and their distinct expression patterns, which potentially contribute to discernible differences in immune function between males and females [65].

## 2. Conclusions

Substantial evidence indicates that gender significantly contributes to the incidence, prognosis, and mortality of many cancers. Although sarcoma remains one of the least studied tumors in terms of sex differences, numerous studies using data collected from the SEER and USSEER databases on a large number of cohorts, and supported by clinical trials, have shown that women also have an advantage over men in sarcoma. Many studies suggest (Table 2) that males have a higher incidence of this type of cancer and are less likely to survive a sarcoma diagnosis.

Significant sex differences have already been observed in childhood malignancies sarcomas. Among the tumors strongly associated with the male sex, childhood sarcomas have been identified as being particularly sensitive to sex biological differences. Differences in incidence between males and females have also been found in different racial and age groups. Although the precise causes behind the disparities in cancer incidence and survival rates between sexes remain unclear, recent research has hinted at the potential involvement of intricate interactions between genetic and environmental factors. These complex interplays might contribute to the generally inferior prognosis observed among males in contrast to females, a trend notable even in soft tissue sarcoma.

Despite this evidence, further research is needed to gain a better understanding of sex differences in sarcoma etiology and prognosis in order to guarantee a therapeutic approach specifically designed for different sexes. In particular, a more systematic approach to examine the disparities in disease rates between male and female patients could be beneficial. This would involve dividing the study into different aspects and categories, such as bone versus soft tissue diseases and pediatric versus adult cases, to confirm the links between translocation and different phenotypes. Supplementing this analysis with fundamental scientific data could help clarify the reasons for these differences.

**Table 2 cancers-16-00201-t002:** Key landmark studies in sarcoma’s gender disparity.

	Study Design, Data Source, Years Include	Relevant Study Population	Key Results
Part A: Evidence for adult			
Rong J et al., 2020 [35]	Retrospective registry-based cohort (CHINA-USA) 2010–2016	1050	Data from gastric GIST patients were collected from the SEER database. Propensity score matching (PSM) was performed to reduce confounding factors, and the clinicopathological features and prognosis of GIST patients were comprehensively evaluated. Gender could be a prognostic factor for gastric GIST survival, and male patients had a higher risk of death.
Mo Chen et al., 2018 [36]	Retrospective registry-based cohort (CHINA-USA) 1973–2013	6582	Data from gastric GIST patients were collected from the SEER database. The study investigated the impact of marital status on the overall survival (OS) and cancer-specific survival (CSS) of operable GIST cases. The marriage could be a protective prognostic factor for survival, and widowed patients had a higher risk of death.
Neal D Freedman et al., 2007 [38]	Retrospective registry-based cohort (USA)	154	The study investigated the association of menstrual and reproductive factors and gastric cancer risk. No associations were observed between gastric cancer risk and age of menarche, number of children, breast feeding, or oral contraceptive use. In contrast, associations were observed with age of menopause, years of fertility, years since menopause, and intrauterine device use.
M Lindblad et al., 2006 [39]	Retrospective (SWEDEN) 1994–2001	612	Esophageal and gastric adenocarcinoma share an unexplained male predominance, A nested case–control study of hormone replacement therapy (HRT) was conducted among 299 women with esophageal cancer, 313 with gastric cancer, and 3191 randomly selected control women. Among 1,619,563 person-years of follow-up, more than 50% reduced risk of gastric adenocarcinoa was found among users of HRT compared to non-users. This inverse association appeared to be stronger for gastric noncardia and weaker for gastric cardia tumors. There was no association between HRT and esophageal adenocarcinoma.
Giun-Yi Hung et al., 2019 [41]	Retrospective registry-based cohort (TAIWAN) 2007–2013	11,393	STS data were acquired from the population-based 2007–2013 Taiwan Cancer Registry of the Health and Welfare Data Science Center, Taiwan. In total, 11,393 patients with an age-standardized incidence rate of 5.62 per 100,000 person-years were identified. Overall, a male predominance and the rate increased with age, peaking at >75 years.
Mei-Chin Hsieh et al., 2013 [42]	Retrospective registry-based cohort (USA) 1995–2008	10,289	STS data were obtained from the North American Association of Central Cancer Registries (NAACCR). The incidence of all STSs combined was higher in males than females.
Rouhani P et al., 2008 [44]	Retrospective registry-based cohort (USA) 1992–2004	12,114	Data from cutaneous soft tissue sarcoma (CSTS) patients were collected from the SEER database confirmed that the incidence of all CSTSs combined was higher in males than females.
Part B: Evidence for childhood and adolescent			
Cole S et al., 2022 [47]	Retrospective registry-based cohort (USA) 1975–2017	5016	Data from osteosarcoma patients were collected from the SEER database. The findings confirm in cases 0 to 9 years old, incidence of primary osteosarcoma was similar between the sexes and increased significantly throughout the study period. Overall, survival rates for all cases have remained relatively unchanged over recent decades, with worse survival observed in males.
Ognjanovic S et al., 2009 [55]	Retrospective registry-based cohort (USA) 1975–2005	987	Data from childhood rhabdomyosarcoma (RMS) patients were collected from the SEER database. The findings revealed the incidence of an embryonal rhabdomyosarcoma (ERMS) was higher in male than females and, more specifically, a smaller peak of embryonal rhabdomyosarcoma (ERMS) incidence rates was observed during adolescence in males which may be related to only those sex-specific hormonal differences.
Ward E et al., 2014 [56]	Retrospective registry-based cohort (USA) 1975–2010	15,780	Data from children and adolescent patients diagnosed with cancer were collected from the SEER database and The North American Association of CentralCancer Registries (NAACCR). The findings confirm that gender disparity has been found in most pediatric cancers, acute lymphoblastic leukemia, non-Hodgkin’s lymphoma, medulloblastoma, hepatic tumors, osteosarcoma, and germ cell tumors, showing that the direct effect of male sex is significant for several tumor types.
Williams LA et al., 2019 [69]	Retrospective registry-based cohort (USA) 2000–2015	71,906	Cancer cases aged 0–19 years were identified using the SEER Program. Male sex was positively associated with most cancers. The higher incidence rates observed in males remained consistent over the childhood and adolescent periods, suggesting that childhood and adolescent hormonal fluctuations may not be the primary driving factor for the sex disparities in childhood cancer. The observed incidence disparities may be due to sex differences in exposures, genetics, or immune responses.
Williams LA et al., 2019 [58]	Retrospective registry-based cohort (USA) 2000–2014	57,004	Cancer cases aged 0–19 years were identified using the the SEER program. Males had worse overall survival and a higher risk of death for acute lymphoblastic leukemia, ependymoma, neuroblastoma, osteosarcoma, thyroid carcinoma, and malignant melanoma.

## Figures and Tables

**Figure 1 cancers-16-00201-f001:**
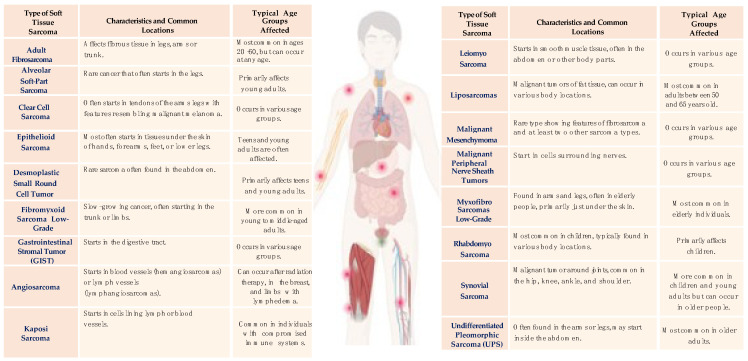
The most common soft tissue sarcomas.

**Figure 2 cancers-16-00201-f002:**
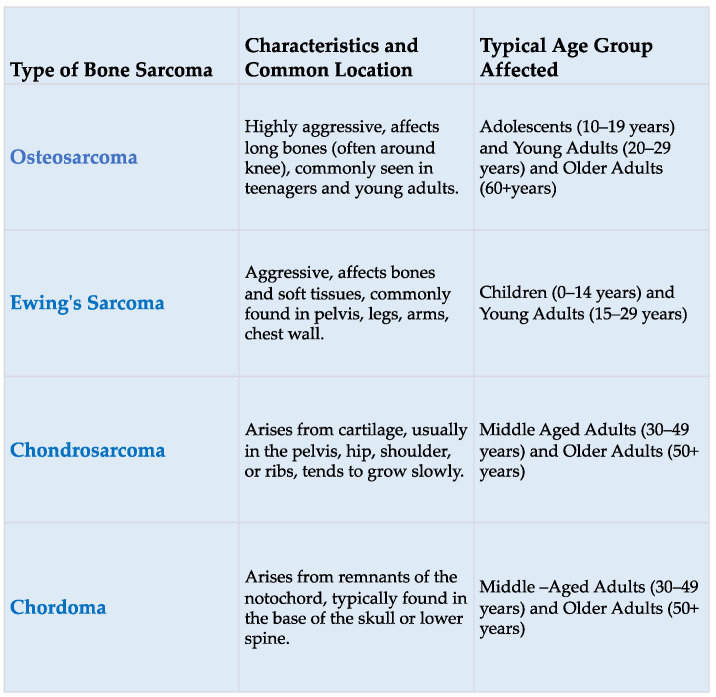
The most common bone sarcomas.

**Table 1 cancers-16-00201-t001:** 2009 Chompret Criteria for germline TP53 mutation screening.

Criterion	Description
I.	Proband with a tumor belonging to the LFS tumor spectrum (e.g., soft tissue sarcoma, osteosarcoma, brain tumor, premenopausal breast cancer, adrenocortical carcinoma, leukemia, and lung bronchoalveolar cancer) before the age of 46 years and at least one first- or second-degree relative with an LFS tumor (except breast cancer if the proband has breast cancer) before the age of 56 years or with multiple tumors.
II.	Proband with multiple tumors (except multiple breast tumors), two of which belong to the LFS tumor spectrum, and the first of which occurred before the age of 46 years.
III.	Patient with adrenocortical carcinoma or choroid plexus tumor, irrespective of family history.
Abbreviations	LFS, Li–Fraumeni syndrome

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
