# Peer review of "Gender Differences in Soft Tissue and Bone Sarcoma: A Narrative Review"

_cancers, 2023, doi:10.3390/cancers16010201_

Round 1

Reviewer 1 Report

Comments and Suggestions for Authors

This was a well-written report that was enjoyable to read.  Some specific comments are included below:

1)     Figure 2: Giant Cell Tumor of Bone is not a sarcoma, and should probably be removed from that figure

2)     Line 66 – the preferred terminology is “radiation-associated” sarcoma, rather than “radiation-induced”

3)     When discussing radiation-associated sarcoma, there is another criterion related to latency of 3 years following radiation.  If including the others, this should be added as well.  Further, given that the thesis of the manuscript is about gender differences, this is a prime opportunity to discuss gender differences in at least the incidence of radiation-associated sarcomas.

4)     The discussion of genetic syndromes associated with sarcoma was outstanding.  However, given that this is a manuscript about gender differences, it is a little awkward that the discussion of gender differences does not begin until Page 7.  Perhaps this should be greatly curtailed to allow for a more robust discussion about gender differences (below), and more inclusion of gender aspects of these syndromes.

5)     Line 364 – is this supposed to read “Sex differences in sarcomas in childhood”?

6)     Line 422-423 – Ewing tumors, Askin tumors, and pNET are now all considered as the “Ewing sarcoma family of tumors”.

7)     The discussion of pubertal influence needs some more depth.  Recognizing that the literature is not terribly robust on the topic, the authors may want to delve into those that arise around adolescence (as suggested).  However, there are childhood sarcomas like ERMS that are typically in prepubescent patients that still exhibit gender inequality in incidence.  Similarly, many sarcomas occur late in life when hormonal differences between men and women are less robust than they are in earlier adulthood.  These nuances should at least be considered and mentioned.

8)     Overall, a more systematic approach to the differences in incidence between male and female patients would be helpful, breaking it down into various aspects and categories – bone vs soft tissue, pediatric vs adult, translocation-associated or not, specific phenotypes, etc..  This would benefit from some augmentation with basic science data to explain why those differences might exist, as well as some more information as to why prognosis may be different.

Author Response

We thank you for all your suggestions, we are sure that they have greatly improved our paper, through your findings now ours has been enriched with the characteristic nuances for such a rare but equally appealing disease .

  • Figure 2: Giant Cell Tumor of Bone is not a sarcoma, and should probably be removed from that figure.

Done as requested we are sorry for the inaccuracy.

  • Line 66 – the preferred terminology is “radiation-associated” sarcoma, rather than “radiation-induced”

Thank you we have corrected

  • When discussing radiation-associated sarcoma, there is another criterion related to latency of 3 years following radiation. If including the others, this should be added as well.  Further, given that the thesis of the manuscript is about gender differences, this is a prime opportunity to discuss gender differences in at least the incidence of radiation-associated sarcomas.

Thank you for pointing out these two incompletenesses, we have incorporated  them with your suggestion in the text indications from line  76 to line 88

  • The discussion of genetic syndromes associated with sarcoma was outstanding. However, given that this is a manuscript about gender differences, it is a little awkward that the discussion of gender differences does not begin until Page 7.  Perhaps this should be greatly curtailed to allow for a more robust discussion about gender differences (below), and more inclusion of gender aspects of these syndromes.

We have reduced the discussion according to his indications.

  • Line 364 – is this supposed to read “Sex differences in sarcomas in childhood”?

Thank you  we have corrected

  • Line 422-423 – Ewing tumors, Askin tumors, and pNET are now all considered as the “Ewing sarcoma family of tumors”.

Thank you we have corrected

  • The discussion of pubertal influence needs some more depth. Recognizing that the literature is not terribly robust on the topic, the authors may want to delve into those that arise around adolescence (as suggested).  However, there are childhood sarcomas like ERMS that are typically in prepubescent patients that still exhibit gender inequality in incidence.  Similarly, many sarcomas occur late in life when hormonal differences between men and women are less robust than they are in earlier adulthood.  These nuances should at least be considered and mentioned.

Thank you for pointing this out to us, we have integrated them with your suggestion in the text from line 468 to line 483.

  • Overall, a more systematic approach to the differences in incidence between male and female patients would be helpful, breaking it down into various aspects and categories – bone vs soft tissue, pediatric vs adult, translocation-associated or not, specific phenotypes, etc.. This would benefit from some augmentation with basic science data to explain why those differences might exist, as well as some more information as to why prognosis may be different.

We agree with your suggestion, but are unable to go into it in depth at this time, so we are keeping it in mind as a future project and mentioning it as a purpose in our conclusions.

Reviewer 2 Report

Comments and Suggestions for Authors

Dear Editor,

I reviewed the manuscript by Cosci ,et al., entitled “Gender differences in Soft Tissue and Bone Sarcoma”

The manuscript is interesting and may provide important information about soft tissue and bone sarcoma. In addition to confirm that males had worse overall survival than females, this study reported worse survival and an increased risk of death for males diagnosed for ependymoma, neuroblastoma, and osteosarcoma. Also, this manuscript is well organized and well written and may be interesting for the reader of the journal. According the manuscript can be published in the present form.

Many thanks

Author Response

Thank you for appreciating our work, we are very happy with your opinion.

Reviewer 3 Report

Comments and Suggestions for Authors

In this narrative review, the Authors aimed to analyze different in sex distribution among bone and soft tissue sarcomas.

The topic is interesting. However, the paper is disorganized.

Please acknowledge the narrative nature of the review in the title and abstract.

Figure 1 and 2: much imprecise information. Age distribution must be detailed.

Actually, it seems that only paragraph 1.4-1.5 and 1.6 is strictly relevant to the topic of the review. Also, these should be further organized in subparagraphs. A table resuming these data might be an added value.

Paragraphs 1.2 and 1.3 are not necessary. The Authors should explain their role in the paper.

Table 3. Not clear.

Author Response

We thank the Reviewer for his/her comments, which improved the quality of the manuscript.

  • The topic is interesting. However, the paper is disorganized.

We have organised the paper a little better by incorporating the suggestions of the reviewers. We hope you will notice the improvement 

  • Please acknowledge the narrative nature of the review in the title and abstract.

Done as requested we are sorry for the inaccuracy.

  • Figure 1 and 2: much imprecise information. Age distribution must be detailed.

Done as requested we are sorry for the inaccuracy.

  • Actually, it seems that only paragraph 1.4-1.5 and 1.6 is strictly relevant to the topic of the review. Also, these should be further organized in subparagraphs. A table resuming these data might be an added value.

We have organised paragraph 1.4 into three subsections, for the other two (1.5, 1.6) we do not see this as possible. Table2. Resume key landmark studies in gender sarcoma’s differences.

  • Paragraphs 1.2 and 1.3 are not necessary. The Authors should explain their role in the paper.

We understand your point about the possible unnecessary nature of the two paragraphs, and on the instructions of Reviewer 1 we have reduced them. We have expalined  closely the authors contribution.

  • Table 3. Not clear.

In the paper, we have provided only two tables.

Round 2

Reviewer 3 Report

Comments and Suggestions for Authors

I really appreciate the Authors' efforts in the attempt to ameliorate their paper. However, they weren't able to reply appropriately to most of my previous concerns. It is not suitable for the publication.

Comments on the Quality of English Language

Many gramma r and syntax errors

Author Response

We are very sorry that we could not reply appropriately to your comments.
We have reviewed and corrected the grammatical and syntax errors as you will be able to see from the highlights.
However, we believe that our manuscript has also improved because of your revision.

Many Thanks
